# WordNet Is All You Need: A Surprisingly Effective Unsupervised Method for Graded Lexical Entailment

**Joseph Renner and Pascal Denis and Rémi Gilleron**

Univ. Lille, Inria, CNRS, Centrale Lille, UMR 9189 - CRIStAL, F-59000, Lille, France
`{firstname.lastname}@inria.fr`

## Abstract

We propose a simple unsupervised approach which exclusively relies on WordNet (Miller, 1995) for predicting graded lexical entailment (GLE) in English. Inspired by the seminal work of Resnik (1995), our method models GLE as the sum of two information-theoretic scores: a symmetric semantic similarity score and an asymmetric specificity loss score, both exploiting the hierarchical synset structure of Word-Net. Our approach also includes a simple disambiguation mechanism to handle polysemy in a given word pair. Despite its simplicity, our method achieves performance above the state of the art (Spearman $\rho = 0.75$) on HyperLex (Vulic et al., 2017), the largest GLE dataset, outperforming all previous methods, including specialized word embeddings approaches that use WordNet as weak supervision.

## 1 Introduction

A crucial aspect of language understanding is the ability to draw inferences between sentences. In many cases, these inferences are directly licensed by the semantics of words: e.g., the sentence *a duck is in the room* entails *an animal is in the room* simply because the concept of duck entails that of animal. These cases of (taxonomic) Lexical Entailment (LE) hold for words whose extensional denotations form a partial order: that is, the set of ducks is included in the set of birds which is itself included in the set of animals.

The taxonomic structure of lexical concepts is a defining aspect of human semantic memory and has been extensively studied in cognitive science as well as in NLP due to its multiple related applications. Initial research milestones include the construction of the WordNet lexical database (Beckwith et al., 2021; Miller, 1995), and the first distributional approaches for automatically detecting hypernym-hyponym pairs (Hearst (1992); Snow et al. (2004); Baroni et al. (2012); Dagan et al.

| X | Y | LE Score |
|---|---|---|
| duck | animal | 5.92 |
| duck | bird | 5.75 |
| conflict | disagreement | 5.20 |
| competence | ability | 4.64 |
| aura | light | 3.69 |
| sofa | chair | 3.38 |
| butter | cream | 2.69 |
| noun | adjective | 0.50 |
| rhyme | dinner | 0.00 |

Table 1: The human lexical entailment scores (0-6) for a small subset of the Hyperlex (Vulic et al., 2017) dataset. Each row should be read as: X entails Y to a degree of LE score.

(2013) *inter alia*). More recently, an important strand of research has led to the development of word representation models that are able to geometrically express asymmetric relations like LE in the embedding space (Roller and Erk, 2016; Vilnis and McCallum, 2015; Nickel and Kiela, 2017).

Inspired by the pioneering works of Rosch (1975) and Kamp and Partee (1995), Vulic et al. (2017) have challenged the traditional view that LE is a binary relation, showing that it is instead a *graded* relation, based on human judgements (i.e., X entails Y to a certain degree). The concomitant release of Hyperlex,[1] a data set of English word pairs scored by humans for the LE relation, has spurred new research into developing models for predicting Graded Lexical Entailment (GLE). A small subset of the dataset is presented in Table 1.

An intriguing research question is whether existing hand-crafted lexical hierarchies like WordNet are indeed able to capture GLE. Preliminary experiments by Vulic et al. (2017); Vulic and Mrksic (2017) report largely negative results: their best WordNet-only based system achieves a mere 0.234 Spearman correlation score with human judgments

---

[1] `https://github.com/cambridgeltl/hyperlex`

from Hyperlex. These poor performance results are blamed on the binary coding of the hypernym-hyponym relation in WordNet. Yet LEAR (Vulic and Mrksic, 2017), the best GLE system to date achieving a 0.682 Spearman correlation score, uses WordNet as a source of constraints for specializing static word embedding models to the task. As static word embeddings alone achieve poor performance for GLE, the question of the contribution of Word-Net in the LEAR improved performance remains open.

In this paper, we propose a simple method that directly and solely exploits the internal structure of WordNet to predict GLE. Our approach relies on Information Content (IC), a continuous information-theoretic measure introduced in Resnik (1995) to model semantic similarity in WordNet. Specifically, we propose to model GLE as a trade-off between a symmetric semantic similarity score and an asymmetric specificity loss score, both of which are defined in terms of IC. Our method is completed with a disambiguation mechanism to address the fact that (G)LE is sense, rather than word specific, and is therefore sensitive to polysemy, an issue that has been largely overlooked in previous work: e.g., the noun *plant* entails *building* only in its working plant sense, and not in its botanical sense. This simple method achieves a 0.744 Spearman correlation score with human judgements, outperforming all previous systems, including specialized word embeddings methods and supervised models, as well as systems based on contextual language models.

To sum-up, our main contributions are threefold. First, we show that the internal structure of Word-Net, as revealed by information-theoretic measures and completed by a disambiguation mechanism, is a reliable predictor of the graded nature of LE. Second, our simple WordNet-only based approach provides a new state-of-the-art for GLE, outperforming previous methods that specialize word embeddings to the LE task using WordNet as weak supervision. Third, we provide a detailed analysis of our method showing the role of the two information-theoretic terms and the importance of sense disambiguation. We also present a simplified version of our scoring function without any frequency information in the computation of IC, which further improves the correlation score (0.753), thus emphasizing the singular importance of Wordnet hierarchical structure for GLE.

## 2 Proposed Method

Given a(n) (ordered) pair of words $(X, Y)$, instantiating a pair of latent (i.e., unknown) concepts $(s_X, s_Y)$, we aim to predict a score $gle(X, Y)$ indicating to what degree $s_X$ entails $s_Y$. Specifically, we propose to compute the score $gle(X, Y)$ as the sum of two terms:

$$gle(X, Y) = Sim(\hat{s}_X, \hat{s}_Y) + SpecLoss(\hat{s}_X, \hat{s}_Y) \tag{1}$$

where $\hat{s}_X$ and $\hat{s}_Y$ are estimations of the latent concepts $s_X$ and $s_Y$. The first term $Sim(\hat{s}_X, \hat{s}_Y)$ stands for a (symmetric) semantic similarity function, capturing the fact that LE requires concepts to be semantically close. The second (asymmetric) term $SpecLoss(\hat{s}_X, \hat{s}_Y)$ encodes another important aspect of LE, namely the fact that there is generally a loss of specificity incurred by using $\hat{s}_X$ (e.g., dog) instead of $\hat{s}_Y$ (e.g., animal), as the set denotation of $\hat{s}_X$ is included in that of $\hat{s}_Y$.[2]

While the general idea of modeling GLE as a trade-off between a similarity term and a specificity term is already present in Vulic et al. (2017), the originality of our approach is to exclusively define these terms using the hierarchical structure of WordNet, a lexical semantic graph made up of word senses (aka *synsets*) and relations between these synsets. This structure is accessed through information-theoretic measures that we define now.

**Information Content (IC)** At first glance, Word-Net might appear inadequate to model GLE because it encodes the hypernym-hyponym relation as a binary relation. But this claim is oblivious of two main facts. First, WordNet has some built-in gradedness as it models the hypernym-hyponym relation as a transitive relation. Second, the binary nature of the taxonomic links in WordNet can be easily bypassed by resorting to the notion of IC. This information-theoretical notion provides a continuous value for synsets by fully exploiting the tree structure associated with the hypernym-hyponym relation. Following Shannon (1951), Resnik (1995) proposes to quantify the information content (aka self-entropy) of each lexical concept $s$ as the log of its inverse probability by $IC(s) = \log(1/P(s))$. While one can simply estimate $P(s)$ via the word frequencies associated with $s$ in a large text corpus, the crucial innovation of Resnik (1995) was to use

---

[2]Synonyms are an obvious exception, as they trivially entail each other while having the same denotation hence specificity.

the taxonomic tree structure of WordNet in this estimation. Specifically, $P(s)$ is estimated as

$$P(s) = \frac{\sum_{h \in Hypo(s)} wc(h)}{\sum_k wc(k)} \quad (2)$$

where $wc(s)$ is the word count for synset $s$ in a large corpus [3] (in our case, Wikipedia), $Hypo(s)$ denotes the set of all hyponym descendants of $s$ ($s$ included), and $k$ stands over all synsets in Word-Net. By fully exploiting the hierarchical structure of WordNet, the notion of IC intuitively captures the monotonic relation between the generality (resp. specificity) of concepts, as measured by their height (resp. depth) in the taxonomy, and their informativeness.

**Similarity**   We define $Sim$ as the IC-based similarity measure introduced in Lin (1998). The similarity between two synsets $\hat{s}_X$ and $\hat{s}_Y$ is defined as the ratio between the information shared by the two concepts, modeled by the IC value of their least common subsumer node (denoted as $lcs$ below), and the information needed to fully describe the two concepts, modeled as the sum of their ICs, leading to

$$Sim(\hat{s}_X, \hat{s}_Y) = \frac{2\, IC(lcs(\hat{s}_X, \hat{s}_Y))}{IC(\hat{s}_X) + IC(\hat{s}_Y)}. \quad (3)$$

**Specificity Loss**   The above similarity measure is arguably a poor predictor of GLE if used alone. This measure will assign high scores to co-hyponyms (e.g., cat and dog) and equal scores to the same hypernym-hyponym pair whatever the order. We therefore need to complement this measure with another, asymmetric measure that is able to quantify the fact that the entailed concept is typically less informative. For this, we define the specifity loss by

$$SpecLoss(\hat{s}_X, \hat{s}_Y) = 1 - \frac{IC(\hat{s}_Y)}{IC(\hat{s}_X)}. \quad (4)$$

This function returns values closer to 1.0 when the $\hat{s}_X$ is more specific than $\hat{s}_Y$ and lower (possibly negative) values when $\hat{s}_Y$ is more specific than $\hat{s}_X$.

The example of co-hyponyms shows the importance of the trade-off between the two scores. Indeed, while the similarity is maximized, the specificity loss is minimized as both synsets have similar

IC values, resulting in a sum that indicates relatively low GLE strength. Similarly, when $\hat{s}_X$ is a *hyper*nym of $\hat{s}_Y$, the similarity score will be high but the specificity score will be low (even negative) and reduce the sum to a more appropriate score.

**Synset Disambiguation**   Turning to the issue of estimating the latent synsets $s_X$ and $s_Y$, we propose to jointly select a pair of synsets with

$$\hat{s}_X, \hat{s}_Y = \underset{s_X \in S(X), s_Y \in S(Y)}{\mathrm{argmax}} Sim(s_X, s_Y) \quad (5)$$

where $S(X)$ and $S(Y)$ denote the set of possible synsets for $X$ and $Y$, respectively. That is, we select the pair of synsets with the maximum similarity value. For example, given the words $plant$ and $building$, this method should hopefully select the synset corresponding to $plant$ as a working plant, not the synset corresponding its botanical sense. We hypothesize that humans implicitly perform such joint sense selection when asked to score the relation between $plant$ and $building$.

## 3   Experiments

This section presents our experimental framework and results of our approach against various baselines and competing systems.[4]

### 3.1   Dataset and Settings

Our evaluation dataset is the Hyperlex dataset (Vulic et al., 2017), which contains 2616 English word pairs (2163 noun pairs and 453 verb pairs). Extracted from WordNet, the pairs from the dataset were scored on a 0-6 scale by human subjects based on the prompt "To what degree is $X$ a type of $Y$?". [5] Scores of the different systems are compared using Spearman's $\rho$ correlation (Spearman, 1904). As our method is fully unsupervised, we can evaluate it and other competing unsupervised methods and baselines on the entire Hyperlex dataset.

For ensuring fair comparison with supervised competitors, we also report the performance of our method on specific test subsets of Hyperlex. Specifically, we rely on the two test subsets provided by the Hyperlex authors: a random subset (25% of the pairs) and a train/validation/test split without any

---

[3] $wc(s)$ is the occurrence count of all words associated with $s$ in WordNet, where a word count is normalized by its total number of synsets.

[4] Our code and data are publicly available at: https://gitlab.inria.fr/magnet/GLE_emnlp.

[5] It is important to note the use of WordNet in creating Hyperlex was restricted to word pair selection, so no structural information from WordNet has influenced the human scores.

lexical overlap (see Vulic et al. (2017) for more details). Finally, note that we use a text dump of Wikipedia for counting word occurrences for IC calculation and frequency baselines.

## 3.2 Unsupervised Systems

**Static Word Embeddings and WordNet Baselines** Our baseline systems are taken or inspired from Vulic et al. (2017). These include a cosine similarity function based on Word2Vec (Mikolov et al., 2013) and the best WordNet-only method reported in Vulic et al. (2017), using the Wu-Palmer similarity (Wu and Palmer, 1994). Finally, Vulic et al. (2017) introduce a strong baseline ($\rho$ score of 0.279) that combines a specificity term, defined in terms of a concept frequency ratio (i.e., $1 - \frac{wc(X)}{wc(Y)}$ for a word pair $(X, Y)$), and a Word2Vec cosine similarity term acting as a threshold.[6] We propose a variation of this approach, by instead summing the Word2Vec vector cosine similarity and the concept frequency ratio. Recall that static embeddings collapse all word senses, thus prevent the use of disambiguation technique in these methods.

**CLM-based Methods** The success of contextual language models (CLM) on many tasks led us to study their usage for GLE. We tested several techniques of deriving static representations from contextual representations following Apidianaki (2023). We found that the best performing one was the method introduced by Misra et al. (2021) (called taxonomic verification) for the related task of graded typicality; in this case, the method uses a GPT-2-XL (Radford et al., 2019) pretrained model. [7] In this approach, taxonomic sentences of the form *"A(n) X is a(n) Y"* are scored by the model, calculating $P(Y|A(n) \ X \ is \ a(n))$. Notice that such contextual prompts allow for some implicit joint disambiguation of the two words.

**Specialized Static Embeddings** The last competitor is the current state-of-the-art LEAR system (Vulic and Mrksic, 2017), which is based on static embeddings specialized for LE through WordNet-derived constraints. Other systems which also use WordNet constraints are Hyper-Vec (Nguyen et al., 2017) and Poincaré Embeddings (Nickel and Kiela, 2017) but their reported performance is lower on the GLE task.

**Comparing Unsupervised Methods** As shown in Table 2, all three baseline systems from Vulic et al. (2017) achieve a $\rho$ score below 0.3. Our baseline combining the concept frequency ratio and a Word2Vec cosine similarity achieves a 0.314 $\rho$ score. Our CLM-based method achieves a 0.425 $\rho$ score which is the best score achieved so far on the GLE task using CLMs. The best competitor to date is the LEAR system with a 0.686 $\rho$ score (taken from Vulic and Mrksic (2017)). [8]

Our WordNet-based method, denoted by WordNet-SSD, reaches a 0.744 $\rho$ score. To our knowledge, this is the best correlation score reported so far on Hyperlex. And it is indeed quite close to the human inter-annotator agreement correlation score of 0.854, which we can take as an upper bound on this task. These results strongly suggest that the hierarchical structure of WordNet provide enough information to accurately model graded LE, and that previous WordNet-based approaches have so far failed at properly leveraging this information.

## 3.3 Supervised Baselines and Competing Systems

We also compare our method's performance to that of the supervised approach presented in Vulic et al. (2017). This method trains a supervised linear regression model on Word2Vec embeddings. As another baseline, we also train a supervised linear regression model using BERT token embeddings, instead of Word2Vec embeddings. Results on the two test splits of Hyperlex are presented in Table 3. The regression model with static embeddings achieves a Spearman's $\rho$ of 0.53 and 0.45 for the random and lexical test splits, respectively, and of 0.420 and 0.257 when using BERT embeddings. On the same splits, our unsupervised method significantly outperforms these supervised models, reaching $\rho$ scores of 0.605 and 0.636, respectively.

## 4 Analysis

This section analyses the different components of our approach via several targeted ablation studies.

---

[6]See Equation (13) in Vulic et al. (2017).

[7]The other pretrained models we tested were bert-base and bert-large (Devlin et al., 2018), roberta-large (Liu et al., 2019), deberta-v3-large (He et al., 2021), and pythia-1b (Biderman et al., 2023).

[8]Note that the system in Wang et al. (2020) is based on the LEAR system, but evaluated the SemEval 2020 English task 2, which is a different (fourth) subset of Hyperlex, achieving a Spearman's $\rho$ of 0.696. We evaluated our method on this subset as well, achieving a Spearman's rho of 0.741.

| Method | all |
|---|---|
| Word2Vec Sim | 0.205 |
| WordNet-Wu-Palmer | 0.234 |
| Frequency ratio with threshold | 0.279 |
| Frequency ratio with sim sum | 0.314 |
| Taxonomic with CLMs (GPT2-XL) | 0.425 |
| LEAR | 0.686 |
| WordNet-SSD | **0.744** |
| Inter-annotator agreement | 0.854 |

Table 2: Spearman's $\rho$ on all Hyperlex word pairs for the methods presented in section 3.2.

| Method | random | lexical |
|---|---|---|
| Linear Reg. (W2V) | 0.530 | 0.450 |
| Linear Reg. (BERT) | 0.420 | 0.257 |
| WordNet-SSD | **0.605** | **0.636** |

Table 3: Spearman's $\rho$ on test splits of the Hyperlex dataset. Note that the first two are supervised, while ours is unsupervised.

**Similarity without Specificity or vice-versa**  We present correlations for the individual similarity and relative specificity functions compared to that of their sum in Table 4. It shows that specificity alone does better than similarity and that the combination does a lot better than any of the two.

**Impact of Disambiguation**  We evaluate our method without synset disambiguation. For this, we do not use Equation 5 for selecting pairs of synsets but instead average the entailment scores over all possible linked synsets pairs in each word pair. The results are reported in Table 5. The last column shows that handling polysemy generally improves correlation scores, while the first three columns further show that the higher polysemy in a pair, the higher the gain.

**Importance of the structure of WordNet**  Our WordNet similarity and specificity loss measures are defined in terms of IC values, which use the tree structure of WordNet to aggregate frequency counts over an auxiliary dataset for synsets. In order to assess the relative importance of the WordNet struc-

| Method | all |
|---|---|
| WordNet Similarity only | 0.393 |
| WordNet Specificity only | 0.521 |

Table 4: Spearman's $\rho$ for our method with $Sim$ only and $SpecLoss$ only.

| Number of synsets | 2-5 | 6-11 | 12+ | All |
|---|---|---|---|---|
| WordNet-SSD-AVG | 0.539 | 0.514 | 0.284 | 0.467 |
| WordNet-SSD | 0.780 | 0.750 | 0.661 | 0.744 |

Table 5: Spearman's $\rho$ over subsets of the Hyperlex dataset along the total number of associated synsets with the word pair where WordNet-SSD-AVG denotes the method with averaging.

| Method | all |
|---|---|
| WordNet-Sim-noFreq | 0.405 |
| WordNet-SpecLoss-noFreq | 0.550 |
| WordNet-SSD-noFreq | **0.753** |

Table 6: Spearman's $\rho$ when using WordNet-SSD with $IC$ without frequency. We also give the scores when using similarity or specificity only.

ture and that of the frequencies in the final correlation scores, we design a new IC measure that does not use word frequencies. Specifically, we simply replaced every word count value with a dummy 1.0 value, in effect assigning uniform probabilities to all synsets. For a synset $s$, its IC value is computed by $IC(s) = -\log(|Hypo(s)|/N)$ where $N$ is the number of synsets in WordNet. Results are shown in Table 6. They show that not using frequency values yields even better correlation scores, furthering our claim that the WordNet structure is essential for GLE.

## 5   Conclusion

We presented a new state of the art method for predicting GLE between word pairs, which solely relies on the structure of WordNet accessed through information-theoretic measures. This new result contradicts previous results arguing that the binary nature of WordNet relations prevented to use WordNet for graded LE. It also shows that a direct use of WordNet performs better than using static word embeddings specialized with WordNet-extracted constraints, suggesting that such methods have not been able to fully leverage the rich structural information of WordNet. Our work also emphasizes the importance of polysemy in this task, an issue being largely ignored in previous work. We tested different CLM-based methods but so far, to the best of our knowledge, no CLM-based method is able to solve the GLE task, thus raising the question whether (enhanced) CLMs are able to model the hierarchical structure of concepts inherent to human semantic memory.

## Limitations

Due to its reliance on WordNet, the proposed approach is currently not applicable to other languages than English. Similarly, the approach is limited in terms of its vocabulary: scoring the lexical entailment of word pairs not covered in WordNet is again not currently possible.

## Ethics Statement

## Acknowledgements

We would like to thank the anonymous EMNLP reviewers for their feedback and suggestions on this paper. This research was funded by Inria Exploratory Action COMANCHE, as well as by the joint IMPRESS project between Inria and DFKI.

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
