# OpenReview forum: "WordNet Is All You Need: A Surprisingly Effective Unsupervised Method for Graded Lexical Entailment"
_EMNLP/2023/Conference — EMNLP 2023 Findings_

### Official Review · Reviewer_6et5 · 2023-08-01

**Soundness:** 3

**Excitement:**

3: Ambivalent: It has merits (e.g., it reports state-of-the-art results, the idea is nice), but there are key weaknesses (e.g., it describes incremental work), and it can significantly benefit from another round of revision. However, I won't object to accepting it if my co-reviewers champion it.

**Paper Topic And Main Contributions:**

This paper proposes a method that outperforms the state-of-the-art of Graded Lexical Entailment (GLE) in English by relying exclusively on WordNet.
The method models GLE as a trade-off between semantic similarity and specificity loss, both defined in terms of Resnik's Information Content.
Results are compared with the state-of-the-art (LEAR) and with those of other related methods.
The paper further reports on ablation studies for analysing the impact of the main components of the method.

**Questions For The Authors:**

Question A: What is the reason for not replicating the experiment with the related methods, using exactly the same data? This would make the results comparable to the state-of-the-art.

**Reasons To Accept:**

Fresh: simple method that outperforms the state-of-the-art of GLE by cleverly relying exclusively on WordNet and on measures that have been here for a long time.

An ablation study that enables to analyse the impact of different components of the method.

**Reasons To Reject:**

Reported results are not completely comparable to those reported for other systems.

The paper does not include actual examples of GLE (including the score).

**Reproducibility:**

4: Could mostly reproduce the results, but there may be some variation because of sample variance or minor variations in their interpretation of the protocol or method.

**Reviewer Confidence:**

4: Quite sure. I tried to check the important points carefully. It's unlikely, though conceivable, that I missed something that should affect my ratings.

**Typos Grammar Style And Presentation Improvements:**

The structure of the paper is not described in the Introduction.

lcs (least common subsumer?) never appears in full.

Examples of the GLE task, including the scores, would help to understand the task at hand.

Results of methods besides LEAR could be included in Table 1.

---

> ### Author Rebuttal · Authors · 2023-08-28
>
> We thank the review for their comments and questions. Please let us know if we missed some of your points or if our comments on the other reviews raised additional questions.
>
> We assure the reviewer our results are directly comparable to the current Hyperlex state of the art unsupervised LEAR system. We also replicated all the unsupervised methods initially presented in Vulic et al. (2017) and obtain similar results, and performed experiments using modern LLMs, the best performing discussed in the results section of the paper, all of which are using the same dataset and evaluation process. Some baselines that performed poorly were not reported in order to respect the 4 pages constraint. With the help of the additional page in case of acceptance, we will make sure to include all these results in Table 1.
>
> However, as mentioned, the system described in Wang et al (2020) achieved the best results on the SemEval 2020 task 2: graded lexical entailment for English. The dataset used for the task and track is a subset of the Hyperlex dataset: part of the dataset was released as a dev set to tune models, while the final results are evaluated on the other (unreleased) subset of Hyperlex, thus making direct comparison with our full Hyperlex results impossible. However, following the reviewer's suggestion, we evaluated our system on the test split of the SemEval task, to allow direct comparison with Wang et al. They report a Spearman's correlation of 0.6963, while our method achieves a Spearman's correlation of 0.7412. Note that their system involves fine-tuning of static embeddings in the attract-repel framework, following the same method as the LEAR system.
>
> Also, the Hyperlex dataset includes 2 train-dev-test splits (split randomly and by lexical properties) for use in supervised settings. We evaluate our unsupervised method on the test sets of these splits, achieving Spearman's correlations of 0.605 and 0.636 for the random and lexical test sets, respectively. The authors of the Hyperlex paper report a maximum Spearman's correlation of around 0.53 and 0.45 for the random and lexical test sets, respectively, using supervised linear regression on static embeddings. We also implemented the same supervised regression models, instead using BERT embeddings, and achieved Spearman's correlations 0.420 and 0.257. We can include these results in a final paper, as they show that even though not a fair comparison (our method is unsupervised and the others are supervised), our method still significantly outperforms.
>
> In case of acceptance, we will include these results in all settings. Also, we will make more precise the GLE task adding examples, include a definition of least common subsumer and describe the structure of the paper in the introduction.

---

### Official Review · Reviewer_rz8W · 2023-08-04

**Soundness:** 2

**Excitement:**

2: Mediocre: This paper makes marginal contributions (vs non-contemporaneous work), so I would rather not see it in the conference.

**Paper Topic And Main Contributions:**

This paper focuses on the task of Graded Lexical Entailment(GLE), which involves predicting the degree  of the lexical entailment relation between two concepts on a continuous scale. The previous approaches for GLE have used distributional semantics and word embeddings, which have limitations in capturing the hierarchical structure of lexical concepts. So this paper proposes a simple unsupervised approach that exclusively relies on WordNet for predicting GLE relations in English. The proposed method models GLE as the sum of two information-theoretic scores: a symmetric semantic similarity score and an asymmetric specificity loss score, both exploiting the hierarchical synset structure of WordNet. As shown in the results of the evaluation experiment, the proposed WordNet-based approach achieves performance above the state of the art on the largest GLE dataset, with a Spearman correlation coefficient of 0.75 on HyperLex. Therefore, authors show that the direct use of WordNet performs better than the use of static word embeddings specialized with WordNet-extracted constraints.


**Questions For The Authors:**

What does the function lcs in Equation 3 represent?

**Reasons To Accept:**

The proposed method outperforms existing methods on the GLE dataset with a Spearman's correlation coefficient of 0.75.


**Reasons To Reject:**

I do not understand the necessity of predicting GLE considering previous studies because there is no explanation of the related studies.
I am not sure if GLE is more useful than LE for capturing transitive lexical entailment relationships such as the dog, animal and living things relationship.
I do not know if the resulting synset pairs can be used for techniques such as word sense disambiguation for polysemous words.


**Reproducibility:**

3: Could reproduce the results with some difficulty. The settings of parameters are underspecified or subjectively determined; the training/evaluation data are not widely available.

**Reviewer Confidence:**

4: Quite sure. I tried to check the important points carefully. It's unlikely, though conceivable, that I missed something that should affect my ratings.

---

> ### Author Rebuttal · Authors · 2023-08-28
>
> We find the reviewer's questions and reasons to reject a bit unclear, but we will respond to the review as we understand it. Please let us know if we missed some of your points or if our comments on the other reviews raised additional questions.
>
> Regarding the necessity of predicting GLE, as mentioned in the paper (see the introduction section), Rosch (1975), Kamp and Partee (1995), and Vulic et al. (2017) have shown the humans think of LE in a graded fashion. In other words, for a certain semantic category, certain member concepts are thought of as more central to the category than others (even while controlling for confounding factors like frequency). Outside of LE directly, this gradedness has implications for NLU tasks such as information retrieval, information extraction, and text summarization, where entailment can be used for better understanding the semantic meaning of text and possible generalization of statements contained therein. We will make sure to make this more clear and develop these ideas more in the paper.
>
> Similarly, please note that the paper should be understood as a short note stating a strong baseline for the GLE task. Indeed, modern language models underperform significantly our method, as we have shown in our BERT baseline and baselines with larger, more recent language models (see our rebuttal to reviewer 1 for more info on these results). Thus, we disagree on the notion that these results are vs. non-contemporaneous work: specialized static embeddings are the current state of the art, even against language models.
>
> Regarding if GLE is more useful than (binary) LE for capturing transitive lexical entailment relationships, both can be used in the exact same manner for predicting transitive relationships, the only difference is GLE gives a level of confidence or strength for the relationships. Furthermore, LE does not differentiate between hypernymy relationships that are more than one level in the concept hierarchy (ie dog -> living thing, dog -> animal are both simply true in LE), while GLE takes this into account when scoring the strengh of a multi-level hypernymy relationship. We will be sure to explain this more thoroughly, but the reviewer should understand that transitive LE relationships are implicit in our discussion of LE vs GLE in the introduction; we make no specific claims about transitive relationships.
>
> Regarding the use of the disambiguation mechanism for word sense disambiguation (WSD), this is beyond the scope of our paper as we never say this mechanism is useful in the different task of WSD on its own, we are just concerned with finding the most relevant pair of synsets between two word pairs in the context of GLE, something that has not been explored in previous GLE studies. WSD (the task on its own) is a different setting in that one needs to find the correct synset of just one word in context.
>
> We disagree with the reproducibility evaluation. We use unsupervised methods and freely distributed LMs/knowledge graphs without any post processing, there is no parameter optimization and all experiments are reproducible using the submitted supplementary materials. This includes raw data, preprocessing code, and exact code for all methods.
>
> We disagree with the soundness evaluation, as there is nothing in the review pertaining to the major technical/methodological problems or insufficient evidence for major claims. Our claims are properly supported by the experiments; perhaps there was a misunderstanding about what our claims are (such as the WSD misunderstanding?).
>
> lcs in equation 3 stands for least common subsumer, which is the synset that is the lowest in the WordNet hierarchy while also being a ancestor node to the two synsets in question. We will explain this further in the text.

---

### Official Review · Reviewer_pJrc · 2023-08-13

**Typos Grammar Style And Presentation Improvements:** 1) Weird sentence at Line no. 249.
**Soundness:** 4

**Excitement:**

4: Strong: This paper deepens the understanding of some phenomenon or lowers the barriers to an existing research direction.

**Paper Topic And Main Contributions:**

This paper addresses the problem of graded lexical entailment. It provides a simple unsupervised approach to the problem, relying solely on the analysis of information found in WordNet. The approach achieves SOTA performance.


**Questions For The Authors:**

A) How justified is the title of the paper ("Wordnet is all you need") when the approach is shown to work only for the GLE task? Can we say that an approach that works well for the GLE task also works well for all NLP tasks?

B) What is "k" in Equation 2?


**Reasons To Accept:**

The paper is well-written.

I think their finding that SOTA results can be attained by merely relying on WordNet's structure is intriguing for the NLP community.


**Reasons To Reject:**

It would have been insightful to observe how well LLMs, like ChatGPT, perform on this task for reference.

They could have easily done this experiment by prompting ChatGPT with the following in the few-shot setting:

"To what degree is X a type of Y?
1) Very low; 2) Low; 3) Medium; 4) High; 5) Very high."

Given the absence of this experiment, the difficulty of the graded lexical entailment problem and the dataset remains unclear.


**Reproducibility:**

4: Could mostly reproduce the results, but there may be some variation because of sample variance or minor variations in their interpretation of the protocol or method.

**Reviewer Confidence:**

3: Pretty sure, but there's a chance I missed something. Although I have a good feel for this area in general, I did not carefully check the paper's details, e.g., the math, experimental design, or novelty.

---

> ### Author Rebuttal · Authors · 2023-08-28
>
> We would like to thank the review for their comments. Please let us know if we missed some of your points or if our comments on the other reviews raised additional questions.
>
> Regarding LLMs, we tested multiple LLMs using a number of methods to score word pairs: two methods using embeddings computed from LLMs, and two methods using the language modeling capabilities of LLMs, using 3 encoder-only language models models: BERT base (2018, 110m parameters), RoBERTa large (2019, 304m parameters),  and DeBERTa-v3 large (2021, 330m parameters). We found the best performing LLM method achieves a Spearman's correlation of 0.361 (mentioned in the baselines section), using a prompting technique from Misra et al (see paper for details) with the BERT model. We did not include all these results and methods due to space. Since reading the review, we also tested 2 additional open-source decoder-only language models: GPT2-XL (2019, 1.5b parameters) and Pythia-1b (2023, 1b paramaters). We found these perform slightly better when using the method introduced in Misra et al, achieving Spearman's correlations of 0.425 for GPT2-XL and 0.394 for Pythia-1b, though still significantly worse and more computationally expensive than our method. We can include all these results in the additional page of space if accepted.
>
> We agree that this could be interesting to use ChatGPT to rank word pairs based on entailment; however, its not entirely trivial on how to extract rankings from ChatGPT. For example, the method suggested in the review (score each word pair on a discrete 1-5 scale) would lead to many word pairs with the same entailment score, which would not be ideal for evaluating using rank correlation. Other options would be to ask ChatGPT to rank on a continuous scale, to give the entire dataset to ChatGPT to rank all at once, or to do a sort of "round-robin" ranking system, developing a full ranking from individual "head-to-head" entailment rankings. We leave these interesting research questions for future work.
>
> Regarding the question A, you are right in that the title ("Wordnet is all you need") is a bit of a overstatement as it is only true for the GLE task. However, there is potential for the use of GLE in other NLP tasks (especially NLU): for example, information retrieval, information extraction, and text summarization could all benefit from understanding to what degree a concept entails another concept. If it's not clear enough though, we can easily revise the title.
>
> In Equation 2, the term k with the summation represents all synsets in WordNet. That is, the denominator is the sum of the word counts for all synsets in WordNet. We will include this explanation in a final version.
>
> In the sentence mentioned in line 249, we meant to say: "Notice that these contextual prompts allow for implicit joint disambiguation of the two words, due to the self-attention mechanism of BERT." We will fix this in a final version.

---

### Meta-Review · Area_Chair_g7uD · 2023-09-19

**Recommendation:** 3

**Metareview:**

This submission proposes to exploit WordNet information to perform graded lexical entailment, showing that a WordNet-based approach reaches high performance on the HyperLex dataset.

While the reviewers appear to diverge quite a bit on their soundness/excitement scores, unfortunately one reviewer failed to perform their duties in acknowledging the author rebuttal, while the authors had raised fair points regarding their review.

The more positive leaning reviewers mention mainly textual aspects, e.g. regarding the somewhat bold title of the paper.

---

### Decision · Program_Chairs · 2023-10-07

**Decision:**

Accept-Findings

**Comment:**

This submission proposes to exploit WordNet information to perform graded lexical entailment, showing that a WordNet-based approach reaches high performance on the HyperLex dataset.

While the reviewers appear to diverge quite a bit on their soundness/excitement scores, unfortunately one reviewer failed to perform their duties in acknowledging the author rebuttal, while the authors had raised fair points regarding their review.

The more positive leaning reviewers mention mainly textual aspects, e.g. regarding the somewhat bold title of the paper.